# Fe-N system at high pressure reveals a compound featuring polymeric nitrogen chains

M. Bykov [1], E. Bykova [1,2], G. Aprilis [3], K. Glazyrin[2], E. Koemets[1], I. Chuvashova [1,3], I. Kupenko[4], C. McCammon [1], M. Mezouar [5], V. Prakapenka[6], H.-P. Liermann[2], F. Tasnádi[7,8], A.V. Ponomareva[8], I.A. Abrikosov[7], N. Dubrovinskaia [3] & L. Dubrovinsky [1]

Poly-nitrogen compounds have been considered as potential high energy density materials for a long time due to the large number of energetic N–N or N=N bonds. In most cases high nitrogen content and stability at ambient conditions are mutually exclusive, thereby making the synthesis of such materials challenging. One way to stabilize such compounds is the application of high pressure. Here, through a direct reaction between Fe and $N_2$ in a laser-heated diamond anvil cell, we synthesize three ironnitrogen compounds $Fe_3N_2$, $FeN_2$ and $FeN_4$. Their crystal structures are revealed by single-crystal synchrotron X-ray diffraction. $Fe_3N_2$, synthesized at 50 GPa, is isostructural to chromium carbide $Cr_3C_2$. $FeN_2$ has a marcasite structure type and features covalently bonded dinitrogen units in its crystal structure. $FeN_4$, synthesized at 106 GPa, features polymeric nitrogen chains of $[N_4^{2-}]_n$ units. Based on results of structural studies and theoretical analysis, $[N_4^{2-}]_n$ units in this compound reveal catena-poly[tetraz-1-ene-1,4-diyl] anions.

[1] Bayerisches Geoinstitut, University of Bayreuth, 95440 Bayreuth, Germany. [2] Photon Science, Deutsches Elektronen-Synchrotron, Notkestrasse 85, 22607 Hamburg, Germany. [3] Material Physics and Technology at Extreme Conditions, Laboratory of Crystallography, University of Bayreuth, 95440 Bayreuth, Germany. [4] Institut für Mineralogie, University of Münster, Corrensstraße 24, 48149 Münster, Germany. [5] European Synchrotron Radiation Facility, BP 220, 38043 Grenoble Cedex, France. [6] Center for Advanced Radiation Sources, University of Chicago, 9700 South Cass Avenue, Argonne, IL 60437, USA. [7] Department of Physics, Chemistry and Biology (IFM), Linköping University, SE-58183 Linköping, Sweden. [8] Materials Modeling and Development Laboratory, National University of Science and Technology 'MISIS', Moscow 119049, Russia. Correspondence and requests for materials should be addressed to M.B.(email: maks.byk@gmail.com)

Since the discovery of a single-bonded cubic nitrogen polymorph (cg-N)[1], many experimental and theoretical studies were dedicated to the search for high energy density nitrogen allotropes and nitrides[2–4]. Polymeric nitrogen solids have been regarded as the best high energy density materials (HEDMs)[5–7] owing to the remarkable difference in the average bond energy between the single N–N bond ($160\,kJ\,mol^{-1}$), the double N=N bond ($418\,kJ\,mol^{-1}$), and the triple N≡N bond ($945\,kJ\,mol^{-1}$)[8]. A number of single-bonded nitrogen allotropes were predicted to exist at pressures higher than the synthesis pressure of cg-N[9–11]. However, in the absence of detailed structural information (e.g., single crystal data) about even those single-bonded nitrogen allotropes which were reported to exist[1,10] any discussion regarding organization of chemical bonding in nitrogen-based HEDMs is difficult. Moreover, it is highly desired to synthesize and stabilize HEDMs at pressures significantly lower than 100 GPa, preferably close to ambient.

Numerous studies suggest that polymeric nitrogen networks may be stabilized at lower pressures in compounds[4,12,13]. Theoretical calculations predict existence of different polynitrides $MN_x$ (M = Li, Be, Na, Mg, Al, K, Ca, Ti, Cr, Rb, Ru, Cs, Hf, W, Re, Os x = 3-10)[14–31] featuring various polymeric nitrogen chains, $N_5$ or $N_6$ rings or even more complex nitrogen networks (e.g., planar $N_{18}$ rings in $KN_8$[23] or $N_{10}$ rings in $BeN_4$[17]). The most straightforward experimental way to obtain these materials could be a direct reaction between a metal, a metal nitride or an azide and nitrogen at high-pressure high-temperature (HPHT) conditions. Previous experiments with metals or metal nitrides and nitrogen in a laser-heated diamond anvil cell led to the synthesis of a variety of transition metal pernitrides $MN_2$ (M = Pt, Ir, Pd, Os, Rh, Ru, Co, Ti) with different structures: Pd and Pt pernitrides have the pyrite-type structure (cubic $Pa$-3)[32,33], $IrN_2$ – the baddelyite-type structure (monoclinic $P2_1/c$)[33], and $OsN_2$, $RhN_2$, $RuN_2$, $CoN_2$—the marcasite-type structure (orthorhombic $Pnnm$)[34–37] and $TiN_2$—$Al_2Cu$-type structure[38]. All of these pernitrides contain dinitrogen N–N units within their structures. Due to the strong covalent N–N bonding, many of these compounds possess exceptionally high bulk moduli suggesting potentially high hardness, which could be further enhanced, if more nitrogen would be incorporated into the structure[31]. For this reason many of the predicted $MN_x$ compounds are often considered not only as HEDMs, but also as possible ultra-hard low-compressible materials[30,31].

In addition, extensive high-pressure investigations of alkali-metal azides ($AN_3$, with A = Li, Na, K or Cs) also aimed at nitrogen polymerization[39–42]. However, unambiguous structural characterization of the obtained high-pressure phases is usually hindered, as they suffer of a lack of crystallinity. Laser heating of cesium azide in a diamond anvil cell (DAC) in the of excess $N_2$ at 60 GPa led recently to the synthesis of a material interpreted as a cesium pentazolate salt $CsN_5$[40]. Shortly after this discovery the first pentazolate-containing complexes were isolated at ambient pressure[43]. This is a good example that information about chemistry and novel bonding of nitrogen at high-pressure may be useful for ambient-pressure synthesis.

The major challenge in the identification of products of HPHT synthesis is the absence of the information on both the chemical composition and the structure. The quality of powder X-ray diffraction data collected in DAC experiments in general is insufficient for solving the structure ab initio. Thus, interpretations of the results are often ambiguous and rely strongly on theoretical predictions.

Here, we overcame this methodological limitation mentioned above. We used laser-heated diamond anvil cells for the synthesis of iron nitrogen compounds through a direct reaction between iron and molecular nitrogen (see Methods for details). The reaction products were characterized using single-crystal X-ray diffraction, and this methodology was extended to over 130 GPa. We report three novel compounds, $Fe_3N_2$, $FeN_2$, and $FeN_4$. The crystal structure of $FeN_4$ possesses polymeric nitrogen chains that are much desired for designing potential high energy density materials. Moreover, our experimental results and theoretical analysis revealed unexpectedly complex chemical bonding in the polymeric nitrogen chains.

## Results and Discussion

**Synthesis and crystal structure of $Fe_3N_2$.** Laser heating of Fe foil in nitrogen medium at 50 GPa and 1900(200) K led to the formation of two nitrides $Fe_3N_2$ and FeN. Iron nitride $Fe_3N_2$ is isostructural to chromium carbide $Cr_3C_2$[44] (Fig. 1a). The structure is built of quadrilateral face-capped trigonal prisms $NFe_7$, which are interconnected by sharing trigonal faces and edges. Such triangular prismatic coordination of six metal atoms about a central nonmetal atom with additional atoms situated outside the quadrilateral faces of the prism is very common for metal-rich compounds containing transition metals and elements with unfilled $p$ levels[45,46]. After laser-heating at pressures above 50 GPa, this phase was no longer observed.

**Synthesis and crystal structure of FeN.** The phase with the chemical composition FeN and the B8 (NiAs) structure type (Fig. 1b) was observed at each pressure-temperature point (Table 1). Very recently, FeN was reported in three independent experimental studies. Clark et al. synthesized NiAs-type FeN by heating $Fe_2N$ in the nitrogen pressure-transmitting medium at ~12 GPa[47], while Niwa et al. have obtained the same compound in a direct reaction between Fe and $N_2$ at ~35 GPa[37]. Our experiments indicate that this compound has a wide stability range and in agreement with a recent study by Laniel et al[48]. On decompression, it is stable down to ambient pressure, but with time, it transforms to zincblende-structured FeN. The volume–pressure dependence for B8-FeN can be described with the third order Birch–Murnaghan equation of state with $V_0 = 34.03(1)\,Å^3$, $K_0 = 185(14)\,GPa$, $K_0' = 6.3(4)$ (Fig. 2a).

**Synthesis and crystal structure of $FeN_2$.** The $FeN_2$ phase was first observed after heating at ~58 GPa. Iron diazenides with different crystal structures ($R\bar{3}m$ and $Pnnm$) were predicted by ab initio calculations[49,50]. We indeed synthesized a phase with the marcasite structure type (space group $Pnnm$, Fig. 1c, Table 2). The structure of $FeN_2$ can be described as consisting of chains of edge-sharing $FeN_6$ octahedra aligned along the c-axis. These chains are interconnected through common vertices. Additional linkage between $FeN_6$ octahedra is provided via N–N bonds (Fig. 1c). According to our structure refinement, the N–N distance at 58.5(5) GPa is 1.307(7) Å, and that is intermediate between the expected bond lengths for double and single-bonded dinitrogen units. For example, the N=N bond in $[N_2]^{2-}$ ions in $BaN_2$ is of 1.23 Å at ambient conditions[51], whereas the calculated N–N bond lengths in $[N_2]^{4-}$ in $PtN_2$ and $OsN_2$ at ambient conditions are 1.41 and 1.43 Å, respectively[52,53].

The compressional behavior of $MN_2$ compounds may give an insight on the oxidation state of the metal and on the bonding between nitrogen atoms. Since, the compression of dinitrides is primarily controlled by the compression of metal-nitrogen (M-N) bonds[34,35,37], the dinitrides with weaker M-N bonds are expected to be more compressible. The strength of a M-N bond depends to a large extent on its ionicity. Therefore, the compressibility of M–N bonds should decrease in the following sequence: $M^{2+}$–N > $M^{3+}$–N > $M^{4+}$–N. This trend is clearly demonstrated by the experimental and theoretical studies. Metals that cannot

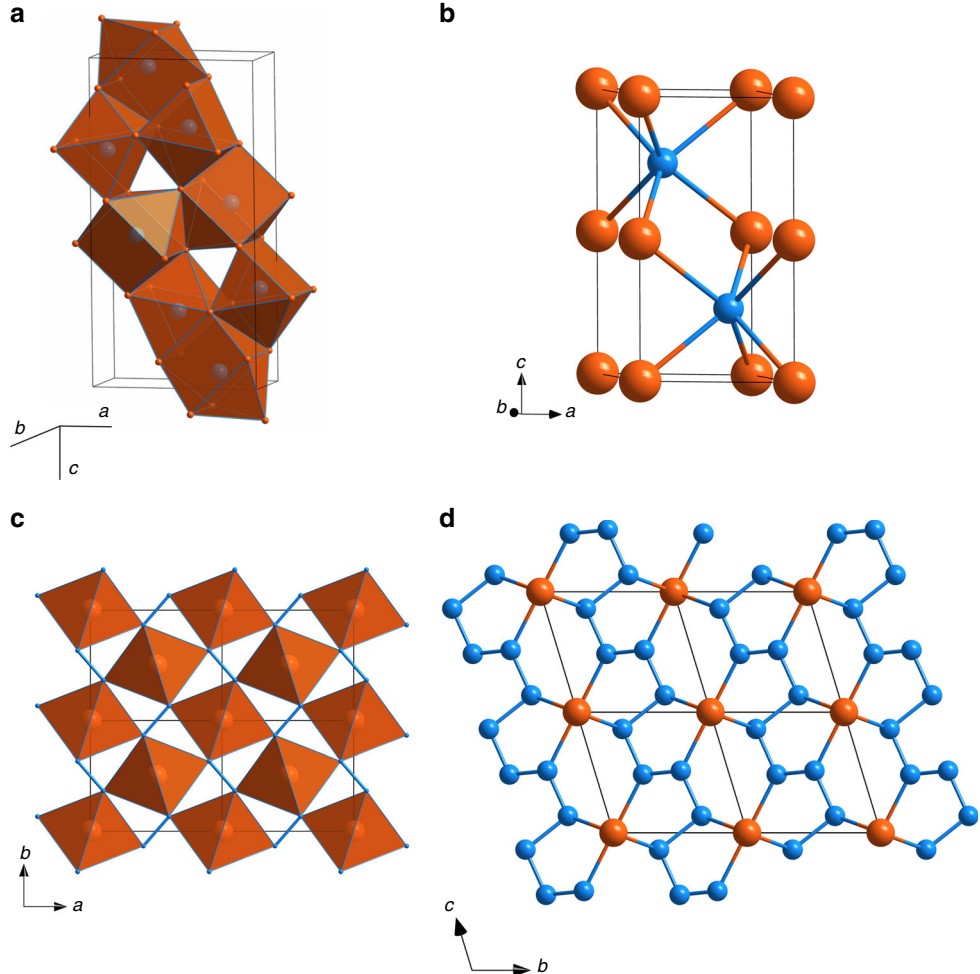

**Fig. 1** Crystal structures of ironnitrogen compounds. Orange and blue balls show the positions of Fe and N atoms, respectively. **a** $Fe_3N_2$ at 50 GPa. The structure is built of quadrilateral face-capped trigonal prisms $NFe_7$, which are interconnected by sharing trigonal faces and edges. **b** FeN at 50 GPa with NiAs structure type. **c** $FeN_2$ at 58 GPa; Shown are the $FeN_6$ octahedra, which are connected into infinite chains through common edges and aligned along the $c$-axis. These chains are interconnected through common vertices. Additional linkage between $FeN_6$ octahedra is provided via N–N bonds. **d** $FeN_4$ at 135 GPa. In the structure of $FeN_4$, each Fe atom is a member of two non-planar five-member $Fe[N4]$ metallacycles, which are almost parallel to the (1-10) lattice plane. Nitrogen atoms form infinite zigzag chains, running along the $c$-direction

**Table 1 Summary of the experimental points at which laser-heating was performed**

| Pressure before heating (GPa) | Pressure after heating | Experiment's number | Temperature (K) | Phases |
|---|---|---|---|---|
| 45.2 | 49.6 | 1 | 1900 ± 200 | $Fe_3N_2$, FeN |
| 55.0 | 58.5 | 1 | 2100 ± 200 | $FeN_2$, FeN |
| 65.1 | 69.6 | 1 | 2200 ± 200 | $FeN_2$, FeN |
| 54.0 | 60 | 2 | >2000 | $FeN_2$, FeN |
| 104[a] | 106.0 | 2 | >2000 | $FeN_4$, FeN |
| 130.0[a] | 135.0 | 2 | >2000 | $FeN_4$, FeN |
| 105[a] | 106.8 | 3 | >2000 | $FeN_4$, FeN |

[a]Pressure estimated by diamond Raman peak[80]

have oxidation state larger than + 2 (Ba, Sr, Ca), form diazenides $M^{2+}[N=N]^{2-}$ with the N=N distances in the range 1.2–1.24 Å[54–56] and rather low bulk moduli ($K_0(SrN_2) = 65$ GPa, $K_0(BaN_2) = 46$ GPa)[51]. Metals that have stable oxidation states + 4 (Os, Ru, Ir, Ti, Pt) form pernitrides $M^{4+}[N-N]^{4-}$ with N–N distances ~1.4 Å and are highly incompressible with very large bulk moduli ($K_0(OsN_2) = 362$ GPa[52], $K_0(IrN_2) = 428$ GPa[57], $K_0(TiN_2) = 385$ GPa[38], $K_0(PtN_2) = 372$ GPa[32]). Regarding the

known pernitrides of those transition metals, that do not readily possess an oxidation state + 4 (Co, Rh), they have intermediate bulk moduli ($K_0(CoN_2) = 216$ GPa, $K_0(RhN_2) = 235$ GPa)[35,37], suggesting the oxidation state of Co and Rh to be +3. According to our data, compressibility of $FeN_2$ could be described with the 3rd order Birch-Murnaghan equation of state with $K_0 = 250(16)$ GPa, $K_0' = 4.0(5)$, and $V_0 = 47.42$ Å$^3$ (Fig. 2b). Therefore, both refined N-N distances and compressibility suggest that Fe in $FeN_2$

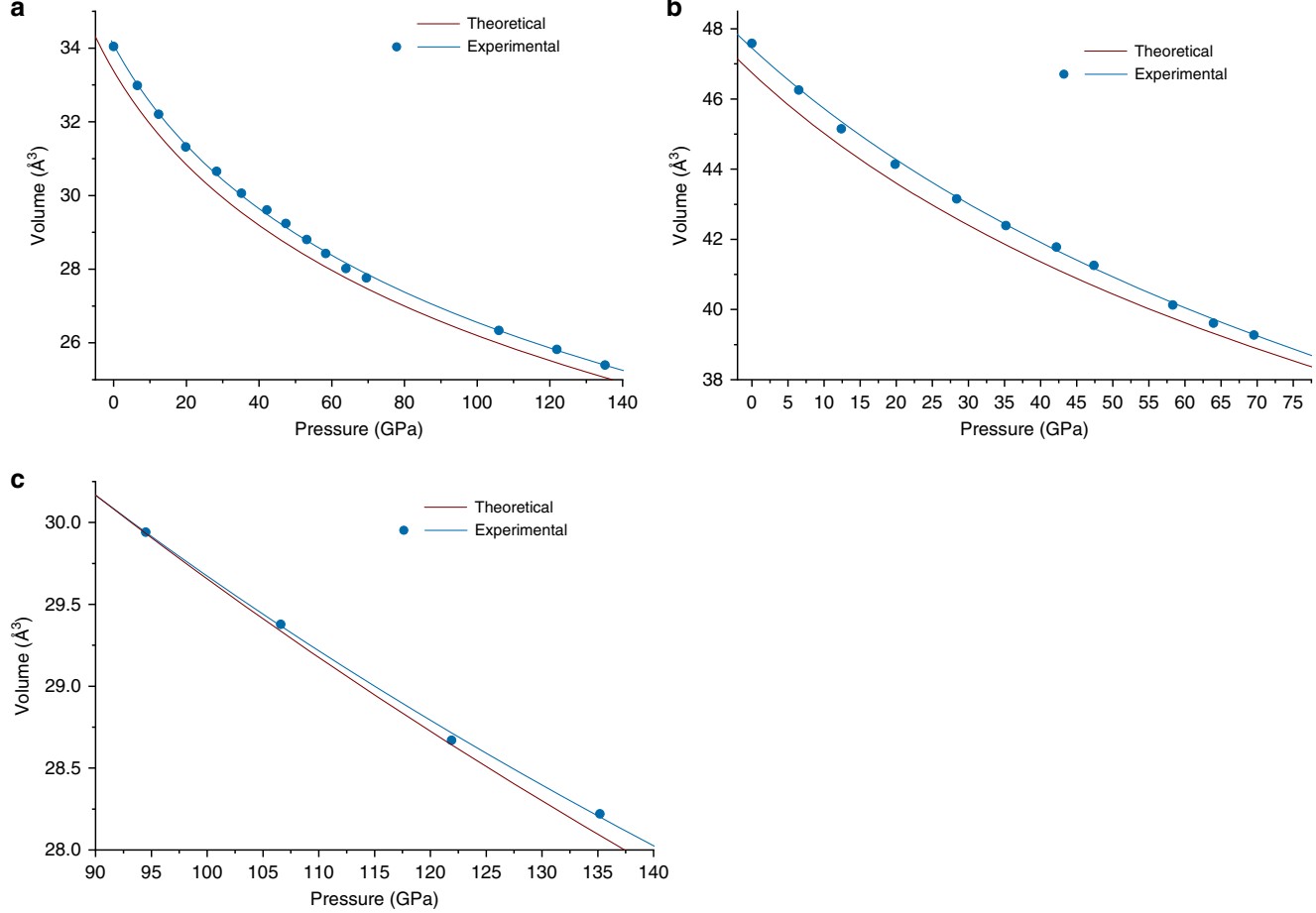

**Fig. 2** Pressure dependence of the unit cell volume of ironnitrogen compounds. **a** FeN, **b** FeN$_2$, and **c** FeN$_4$. Solid curves show the fit of the Birch–Murnaghan equation of state to the experimental data. $V_0(\text{FeN}) = 34.03(1)$ Å$^3$, $K_0(\text{FeN}) = 185(14)$ GPa, $K_0'(\text{FeN}) = 6.3(4)$; $V_0(\text{FeN}_2) = 47.42$ Å$^3$, $K_0(\text{FeN}_2) = 250(16)$ GPa, $K_0'(\text{FeN}_2) = 4.0(5)$; $V_{94.5}(\text{FeN}_4) = 29.94(4)$ Å$^3$, $K_{94.5}(\text{FeN}_4) = 603(22)$ GPa, $K'_{94.5}(\text{FeN}_4) = 4.0$ (fixed)

has an oxidation state + 3. The presence of $[\text{N}_2]^{3\cdot-}$ radical ions is not very likely (however, not excluded)[56], and the electron count in FeN$_2$ can be similar to that suggested for LaN$_2$:[51] Fe$^{3+}$ + $[\text{N}=\text{N}]^{2-}$ + $e^-$, where the electron enters the conduction band.

**Synthesis and crystal structure of FeN$_4$.** The synthesis of FeN$_4$ was first performed at ~106 GPa by laser-heating in nitrogen medium of the mixture of FeN and FeN$_2$, obtained before in the reaction between Fe and N$_2$ at 60 GPa (Experiment #2 in Table 1). The synthesis of FeN$_4$ was reproduced in the Experiment #3 by heating iron foil in nitrogen medium at 106 GPa. In the following discussion, we will always refer to the best-quality single-crystal XRD dataset, which was obtained at ~135 GPa. The indexing of the diffraction pattern resulted in the triclinic unit cell with the parameters $a = 2.5089(4)$, $b = 3.5245(13)$, $c = 3.5409(5)$ Å, $\alpha = 105.08(2)$, $\beta = 110.260(14)$, $\gamma = 92.03(2)°$ (see Table 2, Supplementary Figs. 1–5 and Supplementary Data 4 for details). The crystal structure of the new phase was solved and refined resulting in the composition FeN$_4$ (Figs. 1d, 3). In the structure of FeN$_4$, six nitrogen atoms coordinate each iron atom in the following way: each Fe atom is a member of two non-planar five-member Fe[N$_4$] metallacycles, which are almost parallel to the (1-10) lattice plane (Fig. 3a, b). Two more nitrogen atoms complete the distorted octahedral coordination of Fe (Fig. 3b). The most intriguing feature of the crystal structure is displayed by nitrogen atoms forming infinite zigzag chains, running along the $c$-direction (Figs. 1d, 3).

The geometry of the polymeric nitrogen chains gives an insight into the electron localization within the compound. The N1 atoms have only three neighboring atoms in planar triangular geometry, whereas N2 atoms have tetrahedral coordination (Fig. 3c). This directly suggests the $sp^2$ hybridization of N1 atoms and $sp^3$ hybridization of N2 atoms. Additionally, taking in account the N1–N1, N1–N2 and N2–N2 bond distances, which at 135 GPa equal ~1.29(5), 1.30(3), and 1.43(4) Å, respectively one can classify the N1–N1 bonds as N=N double bonds, and the N1–N2 and N2–N2 bonds as the single bonds. Moreover, theoretical analysis (see below) also shows that double N1–N1 bonds have a significantly higher electron density between atoms than single N1–N2 and N2–N2 bonds (Fig. 3d). Therefore, the nitrogen chains in FeN$_4$ can be considered as catena-poly[tetraz-1-ene-1,4-diyl] anions (Fig. 3e). The tetrazene unit N$_4^{2-}$, thus, serves as a dianionic ligand (Fig. 3e, f), which agrees with the description of Fe atoms in the formal oxidation state +2. The oxidation state +2 is also suggested by the results of Mössbauer spectroscopy (Supplementary Fig. 6). The coordination scheme of Fe atom in FeN$_4$ perfectly matches the 18-electron rule (6 electrons of Fe$^{2+}$ plus 12 electrons from ligands). An attempt to study FeN$_4$ using Raman spectroscopy was not successful due to the strong fluorescence background (Supplementary Fig. 7).

To gain a deeper insight into the bonding features of FeN$_4$, we have performed electronic structure calculations (see Methods). The Bader charge analysis, which shows the charge transfer of $0.37e$ to N2 atoms and $0.25e$ to N1 atoms, is in agreement with the proposed above bonding scheme. For understanding the

**Table 2 Crystallographic data for new iron Fe–N compounds synthesized in the present study at indicated pressures (full crystallographic information is provided in Supplementary Data 1-4)**

|  | Fe₃N₂ | FeN | FeN₂ | FeN₄ |
|---|---|---|---|---|
|  | **Experiment** |  |  |  |
| Pressure (GPa) | 49.6 | 49.6 | 58.5 | 135 |
| Space group | *Pnma* | *P6₃/mmc* | *Pnnm* | P-1 |
| a (Å) | 5.4227(6) | 2.6299(11) | 4.4308(19) | 2.5089(4) |
| b (Å) | 2.6153(3) | 2.6299(11) | 3.7218(11) | 3.5245(13) |
| c (Å) | 10.590(11) | 4.819(7) | 2.4213(18) | 3.5409(5) |
| α(°) | 90 | 90 | 90 | 105.08(2) |
| β(°) | 90 | 90 | 90 | 110.260(14) |
| γ(°) | 90 | 120 | 90 | 92.03(2) |
| V (Å³) | 150.19(16) | 28.86(4) | 39.93(4) | 28.088(13) |
| Z | 4 | 2 | 2 | 1 |
| Fractional atomic coordinates (x/a, y/b, z/c) | **Fe1** (0.4808; 0.25; 0.5996) **Fe2** (0.1282; 0.25; 0.42774) **Fe3** (0.6881; −0.25; 0.7738) **N1** (0.7413; −0.25; 0.9549) **N2** (0.4280; −0.75; 0.7865) | **Fe** (0, 0, 0) **N** (1/3; −1/3; ¼) | **Fe** (0; 0; 0) **N** (−0.4025; 0.1335; 0) | **Fe** (0.5, 0, 0) **N1** (0.160, −0.346, −0.487) **N2** (0.060, −0.303, −0.859) |
|  | **Theory** |  |  |  |
| Pressure (GPa) |  | 45.1 | 55.7 | 134.5 |
| a (Å) |  | 2.62 | 4.45 | 2.49 |
| b (Å) |  | 2.62 | 3.72 | 3.55 |
| c (Å) |  | 4.86 | 2.41 | 3.54 |
| α(°) |  | 90 | 90 | 105.1 |
| β(°) |  | 90 | 90 | 110.4 |
| γ(°) |  | 120 | 90 | 92.1 |
| V (Å³) |  | 28.86 | 39.93 | 28.1 |
| Fractional atomic coordinates (x/a, y/b, z/c) |  | **Fe** (0, 0, 0) **N** (1/3; −1/3; ¼) | **Fe** (0; 0; 0) **N** (−0.4033; 0.1314; 0) | **Fe** (0.5, 0, 0) **N1** (0.149, −0.344, −0.491) **N2** (0.066, −0.312, 0.138) |

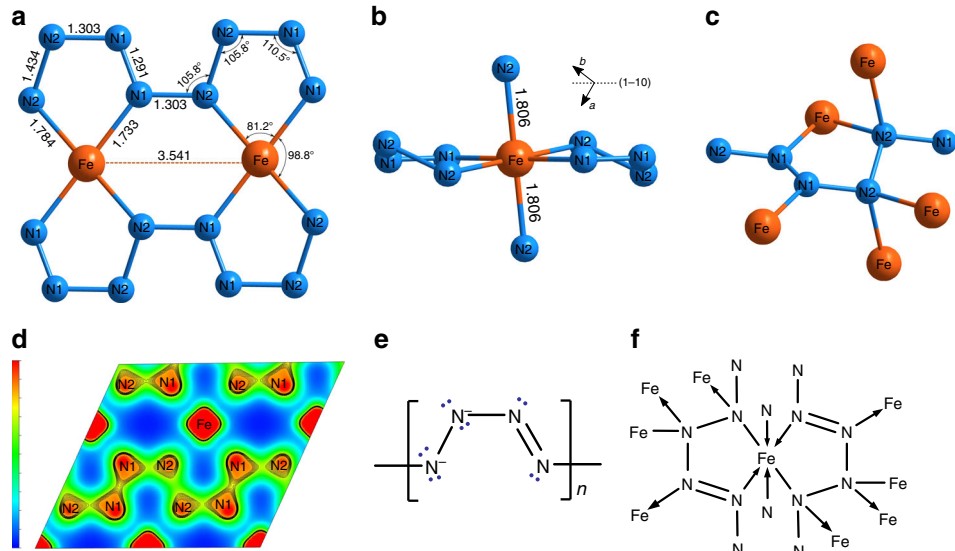

**Fig. 3** Fragments of the crystal structure of FeN₄ at 135 GPa. **a** A fragment of the crystal structure parallel to the (1-10) lattice plane featuring polymeric zigzag N–N chains. Out-off –plane atoms are not shown. **b** The same fragment shown in a different projection. **c** A fragment of the crystal structure showing the coordination geometry of the nitrogen atoms. **d** The charge density map with zig-zag N–N chains in FeN₄ structure. **e** A scheme of poly[tetraz-1-ene-1,4-diyl] anion. **f** A scheme of coordination of iron atoms by poly[tetraz-1-ene-1,4-diyl] anions

dynamical stability of FeN₄, the phonon dispersion relations were calculated at different volumes (Fig. 4). The vibrational frequencies throughout the Brillouin zone are all real, in agreement with the dynamic stability of the phase in the studied pressure range.

We were able to decompress the sample containing FeN₄ obtained in the Experiment #3 down to 22.7(2) GPa (Supplementary Fig. 5); however, the crystal quality significantly decreased and it was not possible to collect a single-crystal diffraction dataset suitable for the reliable indexing and structure

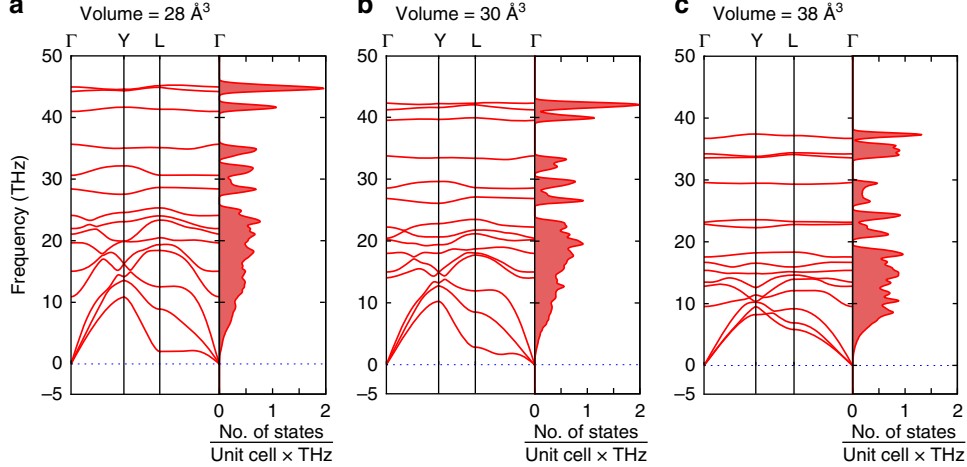

**Fig. 4** Phonon dispersion curves and phonon density of states of $FeN_4$. Calculations were performed with U = 4.0 eV at different volumes, which correspond to pressures of 135 GPa (**a**), 95 GPa (**b**). and 0 GPa (**c**). The phonon density of states are calculated with 0.2 THz smearing

refinement below 94.5 GPa. The pressure dependence of the unit volume data (Fig. 2c) can be described with the second-order Birch–Murnaghan equation of state with the following parameters: $K_{94.5}$ = 603(22) GPa, $V_{94.5}$ = 29.94(4) $Å^3$, where $K_{94.5}$ and $V_{94.5}$ are the bulk modulus and the unit cell volume at 94.5 GPa. The bulk modulus of $FeN_4$ at 94.5 GPa is slightly smaller than that of FeN ($K_{94.5}$(FeN) = 679 GPa). Although the attempt to recover $FeN_4$ was unsuccessful, according to our calculations the compound should be dynamically stable even at ambient pressure (Fig. 4). Therefore, there is still a chance that $FeN_4$ stabilization can be potentially achieved through low-temperature decompression.

Examples of transition metal complexes with tetraazadienes[58], tetraaz-2-enes[59] or hexazene[60,61] ligands are known, and the longest cycle-containing chain has 11 N atoms in 1,1'-(triaz-1-ene-1,3-diyl)bis(1H-tetrazol-5-amine)[62]. However, we are unaware about the existence of linear nitrogen chains containing more than six nitrogen atoms, which have been reported hitherto. Interestingly, polytetrazene-type nitrogen chains were predicted to exist in the high-pressure phase with $N_2H$ stoichiometry, in $BeN_4$ and in $RuN_4$[17,28,63]. In these chains, the ratio of the number of single to double N–N bonds (3:1) is the same as that we found in $FeN_4$, however the structures are different. To the best of our knowledge, $FeN_4$ is the first experimentally confirmed nitrogen compound with polymeric nitrogen chains.

To summarize, in the present work we synthesized a number of compounds in ironnitrogen system, solved and refined their crystal structures, and analyzed their chemical bonding. Our results contribute to both fundamental and applied science starting from fundamental understanding of nitrogen chemistry to the design of novel HEDMs. The first characterized polynitrogen compound is an important milestone for further theoretical and experimental studies. It is reasonable to suggest that compounds like $FeN_4$ may exist in other metal-nitrogen systems and can be metastable down to ambient pressures. Such stabilizaiton can be potentially achieved through low-temperature decompression or through doping. The high synthetic pressure for $FeN_4$ makes it hardly useful for any practical application as a HEDM at present, although its estimated volumetric energy density is 13–15.1 $kJ\,cm^{-3}$ (see Supplementary Methods for details), which is higher than that of TNT (7.2–8.0 $kJ\,cm^{-3}$), 1,3,5-trinitroperhydro-1,3,5-triazine (RDX) (10.1 $kJ\,cm^{-3}$), and pentaerythritol tetranitrate (PETN) (10.6 $kJ\,cm^{-3}$)[25]. However, the information about nitrogen bonding in this compound is

important for further theoretical and experimental studies in the field.

## Methods

**Experiment #1**. Three pieces of iron were loaded into a BX90 diamond anvil cell equipped with 250 µm Boehler–Almax diamonds. The cell was loaded with $N_2$ as a pressure transmitting medium using the gas-loading system installed at the Bayerisches Geoinstitut. Ruby sphere was placed along with the iron pieces for pressure determination. Fe pieces were laser-heated in the DAC at 50, 60, and 70 GPa up to 1900, 2100, and 2200 K, respectively (Table 1) using the portable laser-heating system at the beamline P02.2 at PETRA III[64]. We should note here that generally, pressure in the sample chamber increases after laser heating. Therefore, we provide pressures before and after heating in the Table 1.

**Experiments #2 and #3**. A piece of iron was placed inside a 60 µm hole in a Re gasket, preindented to the thickness of 22 µm. The sample chamber was loaded with nitrogen, which served as a pressure-transmitting medium. We used BX90 diamond anvil cells equipped with Boehler–Almax type diamonds (culet diameter of 120 µm). In the Expreiment#2 the laser-heating was done at 60, 106, and 135 GPa. In the Experiment #3, the sample was first heated at 106.8 GPa. We have used the double-sided laser-heating system of the beamline ID18 of ESRF[64]. In the experiment #2, we have used $^{57}Fe$ as a starting material.

**X-ray diffraction**. The samples were studied by means of single-crystal X-ray diffraction on the synchrotron beamlines P02.2 at DESY, Hamburg, Germany ($\lambda$ = 0.2966 Å, Perkin Elmer XRD1621 flat panel detector); 13IDD at the advanced photon source (APS), Argonne, USA (MAR165 CCD detector, $\lambda$ = 0.2952 Å) and ID27 at ESRF ($\lambda$ = 0.3738 Å, Perkin Elmer XRD1621 flat panel detector). At each pressure step, we collected the X-ray diffraction images upon continuous rotation of the cell from −20° to + 20° ω. At selected pressure points, we collected the data with a narrow 0.5° scanning step in the range from −38° to +38° ω.

In the experiment #1 we determined pressure using the fluorescence line R1 of ruby. In the experiments #2 and #3 we determined pressures using the equations of state of hcp-Fe and/or Re.

Whereas the starting material, a polycrystalline iron foil, gives characteristic Debye-Scherer rings in the diffraction pattern, after the laser-heating in solidified nitrogen, we clearly observed well defined, sharp diffraction spots from multiple grains of new high-pressure phases. Using the $Ewald^{Pro}$ reciprocal space viewing tool for the $CrysAlis^{Pro}$ program[65], we were able to identify the diffraction spots belonging to certain domains, find their orientation matrices and refine the unit cell parameters. The structures of the new phases were solved against single-crystal diffraction data. The general procedure of the analysis of a multigrain diffraction dataset is described in ref. [66]. We provide several raw diffraction images with grain indexing examples in the Supplementary Figs. 1–4 and Supplementary Note 1. Further discussion regarding indexing solutions is given in the Peer Review file.

Diffraction data analysis (peak search, unit cell finding, data integration, frame scaling etc.) was performed with $CrysAlis^{Pro}$ software. The crystal structures were solved using the computer program SHELXT that employs a dual-space algorithm for the solution of a phase problem[67]. General output of the structure solution program was a position of heavy iron atom, while nitrogen atoms were located based on the analysis of residual electron density maps. Crystal structures were refined against single-crystal diffraction data using the computer program JANA2006[68] (see Supplementary Data 1–9 for structural details). The obtained

models represent the authors' optimum refinement of the available X-ray data. The full diffraction data are made available online (see Data availability section). Their complexity for processing is obvious due to the reduced data to parameter ratio, which is characteristic for all single-crystal diffraction data sets obtained in a DAC, and due to the presence of diffraction from numerous domains. If improvements in data processing become available, one can use the present data for reevaluation.

**Calculations.** The ab-initio calculations were performed using the supercell technique and all electron projector-augmented-wave (PAW) method[69] as implemented in the VASP code[70–72]. The simulations were carried out using 4-atoms (B8-FeN), 5-atoms (FeN$_4$) and 6-atoms periodic (FeN$_2$) cells. The integration over the Brillouin zone is performed using the Gamma scheme with $29 \times 29 \times 29$ k-point grids for B8-FeN and $18 \times 18 \times 18$ k-point grids for FeN$_2$ and FeN$_4$ structures. Gaussian smearing method was chosen with a smearing width of 0.05 eV. The energy cutoff for the plane waves included in the expansion of wave functions was set to 500 eV. The convergence criterion for the electronic subsystem was chosen to be equal to $10^{-4}$ eV for two subsequent iterations, and the ionic relaxation loop within the conjugated gradient method was stopped when forces became of the order of $10^{-3}$ eV/Å.

The exchange-correlation energy was described using the Perdew–Wang-91 GGA functional[73] augmented by including Hubbard-U corrections within the DFT + U method following the Dudarev's approach[74]. The chosen parameters U = 4.0 eV and J = 1 eV for the Fe $d$ states provide good agreement with the experimental structural characteristics for all simulated systems FeN, FeN$_2$, and FeN$_4$ structures (Fig. 2, Table 2). We found that FeN is magnetic and used ferromagnetic configuration in our simulations. FeN$_2$, and FeN$_4$ were found to be non-magnetic.

Bader charge analysis[75] derived from topological consideration on the charge distribution was performed using the code developed by Henkelman and colleagues[76] for $400 \times 400 \times 400$ NG(X,Y,Z)F mesh. The phonon calculations were carried out at T = 0 K within quasi-harmonic approximation. We used a finite distortions approach implemented into the PHONOPY program[77] combined with Quantum Espresso (QE)[78] simulations. In the QE calculations, we used plane waves with kinetic energy up to 50 Ry for the electron wave functions while the augmented charges were described using 500 Ry energy cut-off. With these optimized parameters, we reproduced the results of static calculations obtained by VASP. Converged phonon dispersions were achieved using a $(4 \times 4 \times 4)$ supercell with 320 atoms and $(4 \times 4 \times 4)$ Monkhorst-Pack[79] sampling of the Brillouin zone.

**Data availability.** The details of the crystal structure investigations may be obtained from FIZ Karlsruhe, 76344 Eggenstein-Leopoldshafen, Germany (fax: +49-7247-808-666; e-mail: crysdata@fiz-karlsruhe.de) on quoting the deposition numbers CSD-434274—434277. Single-crystal X-ray diffraction dataset for FeN$_4$ at 135 GPa has been deposited to Figshare (https://figshare.com/) with the accession link https://doi.org/10.6084/m9.figshare.6471092.v1. The data that support the findings of this study are available from the corresponding author upon reasonable request.

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

## Acknowledgements

Parts of this research were carried out at the beamline P02.2 at DESY, a member of the Helmholtz Association (HGF). Portions of this work were performed at GeoSoilEnviroCARS (The University of Chicago, Sector 13), Advanced Photon Source (APS), Argonne National Laboratory. GeoSoilEnviroCARS is supported by the National Science Foundation—Earth Sciences (EAR—1634415) and Department of Energy- GeoSciences (DE-FG02-94ER14466). This research used resources of the Advanced Photon Source, a U.S. Department of Energy (DOE) Office of Science User Facility operated for the DOE Office of Science by Argonne National Laboratory under Contract No. DE-AC02-06CH11357. We acknowledge the European Synchrotron Radiation Facility for provision of synchrotron radiation facilities (beamlines ID18 and ID27). N.D. and L.D. thank the Deutsche Forschungsgemeinschaft (DFG projects DU 954-11/1 and DU 393-10/1) and the Federal Ministry of Education and Research, Germany (BMBF, Grant No. 5K16WC1) for financial support. F.T. and I.A.A are grateful to the support provided by the Swedish Research Council project No 2015-04391 and the VINN Excellence Center Functional Nanoscale Materials (FunMat-2) Grant 2016-05156. Support from the Swedish Government Strategic Research Areas in Materials Science on Functional Materials at Linköping University (Faculty Grant SFO-Mat-LiU No 2009-00971) and the Swedish e-Science Research Centre (SeRC) is gratefully acknowledged. Theoretical analysis of structural properties was supported by the Ministry of Education and Science of the Russian Federation (Grant No. 14.Y26.31.0005). Simulations of the electronic structure were supported by the Ministry of Education and Science of the Russian Federation in the framework of Increase Competitiveness Program of NUST "MISIS" (No. K2-2017-080) implemented by a governmental decree dated 16 March 2013, No. 211. Calculations have been carried out at the Swedish National Infrastructure for Computing (SNIC) and at computer cluster at NUST "MISIS". Analysis of the charge density distribution was supported by the RFBR research project No 16-02-00797

## Author contributions

M.B, L.D, N.D., and I.A. wrote the manuscript, M.B., E.B., G.A., E.K., K.G., I.C., M.M., V. P. and H.P.L. performed the experiments, M.B. analyzed and interpreted the diffraction data, I.K. and C. McC. performed and interpreted Mössbauer experiments, F.T., A.P. and I.A. performed the calculations.

## Additional information

**Competing interests:** The authors declare no competing interests.

