## [Peer Review File · Nature Communications]

Reviewers' comments:

Reviewer #1 (Remarks to the Author):

Bykov and co-workers reported the high pressure synthesis of three novel iron-nitrogen compounds (FeN_x). It is interesting that the FeN₄ possesses polymeric nitrogen chains, especially considering its potential applications as potential high energy density materials such as high-performing explosives. However, there are a few technical things that remain unsolved. In the light of the large interest of the materials community on poly-nitrogen compounds, this work should be further considered after major revisions.

1. These new FeN_x compounds (such as Fe₃N₂, FeN₂ and FeN₄) have been thought as promising high energy density materials (HEDMs), however, their energy densities have been not studied. Please evaluate the energy performance (e.g., detonation velocity and pressure) of these FeN_x compounds and compare them with typical high energy density materials such as HMX and CL-20.
2. Of these new FeN_x compounds, FeN₄ is the most promising HEDM due to its highest nitrogen content and an infinite zigzag nitrogen chain. So it is highly desirable to realize stabilization under ambient pressure. In lines 159-161, the authors claimed that there is still a chance FeN₄ stabilization can be potentially achieved through low temperature decompression or through decompression in inert atmosphere. Please try low-temperature decompression and inert-atmosphere decompression experiments, maybe some exciting results could be obtained.
3. In the Introduction, the author claimed that "Their energy density (poly-nitrogen compounds) grows with an increase of the number of catenated nitrogen atoms". In ref. 2 and 49, some longer catenated nitrogen chains (e.g., cyclopentazolate or N₁₁ chain) were cited. Does this mean these reported chain compounds has much higher energy level than these FeN_x compounds?
4. The in situ single-crystal synchrotron X-ray diffraction was used to reveal the crystal structure of novel iron-nitrogen compounds. What about the in situ spectroscopic techniques, e.g., in situ FTIR and Raman? Please supplement spectroscopic experiments.
5. The crystal data need to be further refined and the Checkcif files of three novel iron-nitrogen compounds should be supplemented.

Reviewer #2 (Remarks to the Author):

This a simple, straightforward paper with a strong result. Synthesis of FeN₄ with polymeric nitrogen chains is an exciting discovery.

Issues:

1. Line 38-39 "Moreover, it is highly desired to synthesize and stabilize HEDMs at pressures significantly lower than 100 GPa, preferably close to ambient."

Of course this is true and everybody says it. The problem I have with it is that significantly lower pressures also imply significantly less work done per unit mass to achieve the lower pressure. Therefore, unless another energy source is available, the synthesized compound cannot be as energetic as it would have been had it been made at high pressures. We always have to ask: if we're making an energetic compound where is the energy coming from?

On the other hand any compound that requires more than around 10 GPa is never likely to be useful for any application as an HEDM (and 106 GPa is about as bad as cg-N). I hope that it is

possible to achieve a compromise in which "fairly" energetic materials can be made under only mildly "extreme" conditions. Otherwise I see no future for this line of research other than for pure scientific interest.

So I'd like the authors to address this concern in the introduction. I'm obviously not expecting them to somehow solve the problem here, but they need to be clear about the potential negative implications of reducing synthesis conditions.

2. The authors have used only a single experimental diagnostic as far as I can tell. That's fine if that diagnostic is definitive. However, unless I missed it they provide no raw diffraction patterns. They absolutely must do that. Use supporting materials if necessary but have at least some data in the main text.

3. Minor: why iron? Is there any particular motivation?

Reviewer #3 (Remarks to the Author):

The manuscript claims the formation of three new iron-nitrogen phases (Fe_3N_2 , FeN_2 and FeN_4), formed by reaction at high pressure and temperature in a diamond anvil cell and probed by single-crystal X-ray diffraction. The contribution of three new phases is impressive. Two of them (Fe_3N_2 and FeN_2) have known structural analogues (Cr_3N_2 and marcasite). The most interesting aspect of the work is the claim of novel triclinic FeN_4 structure that possess polymeric nitrogen chains. While the work should certainly be published (although perhaps in a more specialized journal), it would require major revision for *Nat. Commun.* as many critical aspects are missing / unclear and the evidence is not entirely convincing.

It is expected that the authors would provide some characterization beyond an incomplete crystal structure of FeN_4 that supports the conclusion of polymeric N chains. While the reviewer sympathizes with the difficulty in obtaining single-crystalline data sets at high-pressure, further characterization methods would enhance the study. Raman or IR, for example, could provide additional evidence for the formation of this phase. While the authors say that many groups rely too heavily on DFT calculations, a structure prediction calculation could also provide valuable support to compliment the limited diffraction data set.

There are not sufficient details for the structure solution for the FeN_4 phase. In general, it seems unlikely that a robust indexing of the P-1 phase FeN_4 was possible with the limited number of reflections used in the refinement (49), even with such a small unit cell. Did the authors use a theoretical model to support this indexing initially or complete the indexing without prior theoretical support? How was the quality of the indexing assessed? The cif file says that the structure solution was obtained with SHELXT, but no details are given. An explicit ordering of methods in the manuscript would make it clearer to the reader as to the veracity of the structural solution for FeN_4 .

What is the explanation for the significant difference in bond distances for N1-N2 and N2-N2 (1.33 and 1.43 Å) if they are both considered as N-N single bonds in FeN_4 ? Increased electron density would certainly open up the N2-N2 distance, but 1.43 is the longest known pernitride distance! In this case, the π^* orbitals are fully occupied (N_2^{4-}) and it's not clear how this could be consistent with the current result.

Regarding the presence of infinite N chains, it is unclear, based upon the presented data and bond distances, to the reviewer that the chains are really infinite and not the result of some unmodelled disorder or weak(er) interaction between discrete $(\text{N}_4)^{2-}$ anions. The R-factor of the best data set for FeN_4 is significantly worse than the other presented data sets, $R_1 = 0.0879$, and the N atoms are not refined anisotropically while they are in the others. While this is to be expected from a low-

symmetry material, the determination of the atom positions and their interactions can certainly be improved. This would help elucidate the presence of infinite chains vs discrete anions/disorder. The authors point out that the best single-crystal refinement of FeN₄ was obtained at ~135 GPa, but that the phase persists on decompression down to 22.7 GPa. Although they were unable to obtain single-crystal refinements below 94.5 GPa, it should be possible to provide the lattice indexing for data collections below 94.5 GPa. This would allow for additional data points for the determination of the EOS beyond the small number presented (4).

The phase FeN was present throughout the study. What was the phase fraction of FeN during the various points in the study? Are the authors able to isolate crystalline regions of a single phase or is each other phase always mixed with FeN? If not, what methods are used to deconvolute the presence of multiple single-crystalline phases? What is the reason(s) for the large range of stability presented by FeN?

The authors seem to claim that FeN₂ is a pernitride on page 4, but the bond length and compressibility seem to rule this out. The Fe³⁺ interpretation seems plausible, but the authors might consider this as a diazenide with an elongated N=N bond due to the extra electron in the conduction band.

The authors should consider revising the sentence "High nitrogen content and stability are mutually exclusive, thereby making the synthesis of such materials challenging." Of course pure nitrogen phases (like cg-N) can be thermodynamically stable depending on conditions. What the authors are really referring to here is metastability.

It is not clear why the authors separate chain and ring structures in the discussion of polymeric nitrogen phases in the introduction. Much progress has been made in pentazolate synthesis recently and several additional references (papers by Steele in ref. 30, for example) could be expanded.

TiN₂ has the Al₂Cu structure, not marcasite as stated on page 3.

Reviewer #4 (Remarks to the Author):

This paper reports the extreme conditions synthesis of several new Fe-N systems, the authors should be congratulated on the successful application of very challenging experimental conditions resulting in novel findings.

The topic of high energy density materials is topical and attractive to a wide range of readers, both in the direct community and the wider scientific community.

The majority of the conclusions seem well founded, however certain aspects of the paper would benefit from either further explanation or investigation. Additionally there are minor typographical errors which detract from the readers flow through the paper. Concerns about the detailed methods and conclusions are outlined below with additional comments on the typographical errors later in this report.

From the current description on the synthesis of the materials, a noted challenging field, the details do not clarify the conditions suitably for reproduction of the experiment. The laser heating is described as happening at 50GPa at 1900(100)K, the error on the pressure here is difficult to obtain, but more pertinently the details of if this is the loading pressure or the maximum pressure obtained should be described. If this is the loading pressure then the reaction probably occurs as a pressure that is far in excess of this by impact of the extreme heating of the sample. Additionally the details of the pressure measurements are not entirely clear and the inclusion of all parameters

for this, for instance: if this is a measurement from Ruby fluorescence then have the authors accounted for the lack of linearity in the pressure response due to the pressure range investigated as well as the effects of temperature?

On page 4 the FeN system is described as transforming back to the ZnS form, without detailing which ZnS motif is being mimicked.

A large uncertainty in the paper, in its current form, is the interpretation of the bonding of the N₂ moieties described on page 6. Comparisons of N₂ bonding distances under ambient conditions to those at 135GPa is questionable when used for the determination of multiplicity of bonding. Why not compare to data from higher pressure studies? Moreover within this section the value of 1.33 Å is described as being singly bonded (the same as 1.22Å whereas 1.43Å is described as doubly bonded), from the values given 1.33 falls directly between them. This midpoint is further backed by the calculated electronic properties, whereby a simple description of this bond as single should be strongly interrogated.

The assertion on page 7 that the compound may be recovered to ambient conditions through decompression under inert atmosphere would benefit from further discussion, the sample currently is shown to degrade at 22.7(2)GPa (note there is error inclusion here) however under those conditions the sample MUST surely still be maintained in an inert atmosphere hence the implication that the atmosphere is causing the degradation is somewhat flawed.

With suitable care and attention all of the above could be actioned.

The following relate to typographical issues with the manuscript in its current form:

Page 1: chromium carbide Cr₃N₂ is incorrect

Page 2 and possibly elsewhere referencing inconsistencies 3-5 and then 10,11,12

Page 2: paragraph 2 "as of today" reword ", as yet," ?

Page 3 DAC is not defined prior to use - very common usage but to the wider audience this may be lost

Page 3 paragraph 2 "Major challenge" word missing add either "The" or "A" at start of sentence, "rely too strongly" loose english "is reliant upon" or "relies strongly"

Page 4 1st paragraph "we were no longer observing this phase" again loose replace with "this phase was no longer observed"

Page 7 spelling of stabilization

Page 8 referencing inconsistency period after and before reference numbers

Page 10 missing space between were and performed

We are grateful to all Reviewers for a detailed inspection of our work, for pointing out the importance of the research, and for their comments that helped us to significantly improve the quality of our manuscript. We have extended our manuscript by the supplementary materials. In the revised version of the manuscript all changed sentences are marked yellow. Below we provide point-by-point answers to the Reviewers' comments.

Reviewer #1 (Remarks to the Author):

Bykov and co-workers reported the high pressure synthesis of three novel iron-nitrogen compounds (FeN_x). It is interesting that the FeN₄ possesses polymeric nitrogen chains, especially considering its potential applications as potential high energy density materials such as high-performing explosives. However, there are a few technical things that remain unsolved. In the light of the large interest of the materials community on poly-nitrogen compounds, this work should be further considered after major revisions.

=====

Comment 1:

These new FeN_x compounds (such as Fe₃N₂, FeN₂ and FeN₄) have been thought as promising high energy density materials (HEDMs), however, their energy densities have been not studied. Please evaluate the energy performance (e.g., detonation velocity and pressure) of these FeN_x compounds and compare them with typical high energy density materials such as HMX and CL-20.

Reply 1:

Inspired by Reviewer's request we have performed the calculations of the energy density of FeN₄ and included them into the supplementary materials (Page 9).

Changes in the main text:

We add a short discussion on the page 7 of the manuscript: "its estimated volumetric energy density is 13 - 15.1 kJ/cm³ (see supplementary materials for details), which is higher than that of TNT (7.2-8.0 kJ/cm³), 1,3,5-trinitroperhydro-1,3,5-triazine (RDX) (10.1 kJ/cm³), and pentaerythritol tetranitrate (PETN) (10.6 kJ/cm³)"

=====

Comment 2:

Of these new FeN_x compounds, FeN₄ is the most promising HEDM due to its highest nitrogen content and an infinite zigzag nitrogen chain. So it is highly desirable to realize stabilization under ambient pressure. In lines 159-161, the authors claimed that there is still a chance FeN₄ stabilization can be potentially achieved through low temperature decompression or through decompression in inert atmosphere. Please try low-temperature decompression and inert-atmosphere decompression experiments, maybe some exciting results could be obtained.

Reply 2:

We agree with the Reviewer that some exciting results might be obtained if FeN₄ could be isolated at ambient pressure. However, the main goal of our study was the synthesis of this compound, while its isolation and purification is beyond the scope of our work. We would like to point out that sometimes the development of a methodology of certain high pressure material production at ambient pressure requires joint efforts of dozens of groups and takes years (high pressure boron polymorphs is one of such examples), but sometimes the information that a certain compound exists at least at high pressures leads to almost immediate synthesis of the same compound using a completely different route. We are convinced that publication of our work will stimulate search for alternative ways to synthesize/recover iron polynitrides.

=====

Comment 3:

In the Introduction, the author claimed that "Their energy density (poly-nitrogen compounds) grows with an increase of the number of catenated nitrogen atoms". In ref. 2 and 49, some longer catenated nitrogen chains (e.g., cyclopentazolate or N₁₁ chain) were cited. Does this mean these reported chain compounds has much higher energy level than these FeN_x compounds?

Reply 3:

We thank the Reviewer for this comment. "Energy density" is a term that can be used in different ways, e.g. one can mean volumetric or gravimetric energy density, or energy density per molecule. The statement in the original sentence was valid for energy density per molecule, i.e. a molecule containing more N-N bonds potentially has higher energy density. This is clarified in the revised version of the manuscript.

Changes in the main text:

We rephrase the first sentences in abstract (Page 1) in the following way: "Poly-nitrogen compounds have been considered as potential high energy density materials for a long time due to the large number of inherently energetic N-N or N=N bonds. In general, the higher nitrogen content leads to a higher energetic performance, however in most cases high nitrogen content and stability at ambient conditions are mutually exclusive, thereby making the synthesis of such materials challenging".

=====

Comment 4:

The in situ single-crystal synchrotron X-ray diffraction was used to reveal the crystal structure of novel iron-nitrogen compounds. What about the in situ spectroscopic techniques, e.g., in situ FTIR and Raman? Please supplement spectroscopic experiments.

Reply 4:

In our work we use single-crystal X-ray diffraction as a primary and most powerful method for the determination of the crystal structure. This method provides unambiguous results, which do not require any additional confirmation using other techniques. We have, however, crosschecked the results using *ab initio* calculations, which fully support our conclusions and provide additional and more detailed information about chemical nature and properties of Fe-N compounds. Following the suggestion of the Reviewer, we have included several Mössbauer and Raman spectra of FeN₄ to the supplementary materials (Pages 7 and 8).

Changes in the main text:

We mention the results of Mössbauer spectroscopy with a reference to the supplementary materials on the page 6 of the revised manuscript.

=====
=====

Comment 5:

The crystal data need to be further refined and the CheckCif files of three novel iron-nitrogen compounds should be supplemented.

Reply 5:

We include checkcif reports for all phases in the revised version of the manuscript. We were also able to improve the refinement of FeN₄ (see also reply to the Comment#4 of the Reviewer #3).

Changes in the main text:

Slightly changed lattice parameters and atomic coordinates of FeN₄ (Page 5, Figure 2, Table 2).

=====
=====

Reviewer #2 (Remarks to the Author):

This a simple, straightforward paper with a strong result. Synthesis of FeN₄ with polymeric nitrogen chains is an exciting discovery.

=====

Comment 1:

Line 38-39 "Moreover, it is highly desired to synthesize and stabilize HEDMs at pressures significantly lower than 100 GPa, preferably close to ambient."

Of course this is true and everybody says it. The problem I have with it is that significantly lower pressures also imply significantly less work done per unit mass to achieve the lower pressure. Therefore, unless another energy source is available, the synthesized compound cannot be as energetic as it would have been had it been made at high pressures. We always have to ask: if we're making an energetic compound where is the energy coming from?

On the other hand any compound that requires more than around 10 GPa is never likely to be useful for any application as an HEDM (and 106 GPa is about as bad as cg-N). I hope that it is possible to achieve a compromise in which "fairly" energetic materials can be made under only mildly "extreme" conditions. Otherwise I see no future for this line of research other than for pure scientific interest.

So I'd like the authors to address this concern in the introduction. I'm obviously not expecting them to somehow solve the problem here, but they need to be clear about the potential negative implications of reducing synthesis conditions.

Reply 1:

We agree with the Reviewer and stated on the bottom of the page 7 that the high synthetic pressure for FeN₄ makes it hardly useful for any practical application. Still, the information about nitrogen bonding in this compound is important for further theoretical and experimental studies in the field, and we cannot exclude that other compounds (or other methods of synthesis) may be very useful at the end.

Changes in the main text:

The additional sentence (Page 7) is formulated as follows: The high synthetic pressure for FeN₄ makes it hardly useful for any practical application as a HEDM at present, although its estimated volumetric energy density is 13 - 15.1 kJ/cm³ (see supplementary materials for details), which is higher than that of TNT (7.2-8.0 kJ/cm³), 1,3,5-trinitroperhydro-1,3,5-triazine (RDX) (10.1 kJ/cm³), and pentaerythritol tetranitrate (PETN) (10.6 kJ/cm³). However, the information about nitrogen bonding in this compound is important for further theoretical and experimental studies in the field.

=====

Comment 2:

The authors have used only a single experimental diagnostic as far as I can tell. That's fine if that diagnostic is definitive. However, unless I missed it they provide no raw diffraction patterns. They

absolutely must do that. Use supporting materials if necessary but have at least some data in the main text.

Reply 2:

Unfortunately, raw single-crystal diffraction data for our experiments have a total size of 141.15 GB. Nature Communications has a limit of 150 MB for all supplementary information, which is about 1000 times less than is necessary. A single dataset has a size of ~ 2GB: it consists of 152 16-bit .tif images, that are almost useless without each other.

The problem of raw data deposition is widely discussed in the crystallographic community (see, for example Ref. [1]) However, to date, it is accepted, that a crystallographic information file (cif) with a relevant checkcif report is a sufficient proof of the structure determination. As a matter of fact, specialized crystallographic journals (Acta Crystallographica, Zeitschrift für Kristallographie) never ask for raw datasets.

In the revised manuscript, we add several examples of the diffraction images to the supplementary information (Pages 2-4) showing how diffraction peaks from different phases look like and that they are well separated from each other, allowing standard data integration software to be used. We believe that these images may especially be useful for those readers, who work with laser-heated diamond anvil cells; so that they can recognize that their data are suitable for single-crystal diffraction analysis too.

In the revised version of the manuscript we also include the checkcif reports.

Finally, it is no need to say that any raw diffraction data mentioned in the paper will be accessible upon request (as required by Nature group publication policy)

Changes in the main text:

We add the following sentence at the page 9: "We provide several raw diffraction images with grain indexing examples in the Supplementary Information file (Supplementary figures 1-3)."

=====
=====

Comment 3:

3. Minor: why iron? Is there any particular motivation?

Reply 3:

We will be honest here. Our initial motivation was to study iron-nitrogen system at conditions relevant for the Earth Core. However, the results, that we obtained, led us away from immediate geoscientific applications.

=====

=====

Reviewer #3 (Remarks to the Author):

The manuscript claims the formation of three new iron-nitrogen phases (Fe₃N₂, FeN₂ and FeN₄), formed by reaction at high pressure and temperature in a diamond anvil cell and probed by single-crystal X-ray diffraction. The contribution of three new phases is impressive. Two of them (Fe₃N₂ and FeN₂) have known structural analogues (Cr₃N₂ and marcasite). The most interesting aspect of the work is the claim of novel triclinic FeN₄ structure that possess polymeric nitrogen chains. While the work should certainly be published (although perhaps in a more specialized journal), it would require major revision for Nat. Commun. as many critical aspects are missing / unclear and the evidence is not entirely convincing.

=====

=====

Comment 1:

It is expected that the authors would provide some characterization beyond an incomplete crystal structure of FeN₄ that supports the conclusion of polymeric N chains. While the Reviewer sympathizes with the difficulty in obtaining single-crystalline data sets at high-pressure, further characterization methods would enhance the study. Raman or IR, for example, could provide additional evidence for the formation of this phase. While the authors say that many groups rely too heavily on DFT calculations, a structure prediction calculation could also provide valuable support to compliment the limited diffraction data set.

Reply 1:

Apart from crystal structure determination from single-crystal X-ray diffraction, in the original version of the manuscript we provided the results of the *ab initio* calculations that fully support the experiment. Furthermore, we presented phonon dispersions, and charge density maps. We would like to point out here that our calculations do not aim at complimenting the limited diffraction data set (which is in fact fully sufficient for the structural solution and refinement), but provide an independent cross-verification of the results.

In the supplementary information of the revised version of the manuscript we also provide results of Mössbauer and Raman spectroscopies (Pages 7, 8).

Changes in the main text:

We mention the results of Mössbauer spectroscopy with a reference to the supplementary materials on the page 6 of the revised manuscript.

=====

=====

Comment 2:

There are not sufficient details for the structure solution for the FeN₄ phase. In general, it seems unlikely that a robust indexing of the P-1 phase FeN₄ was possible with the limited number of reflections used in the refinement (49), even with such a small unit cell. Did the authors use a theoretical model to support this indexing initially or complete the indexing without prior theoretical support? How was the quality of the indexing assessed? The cif file says that the structure solution was obtained with SHELXT, but no details are given. An explicit ordering of methods in the manuscript would make it clearer to the reader as to the veracity of the structural solution for FeN₄.

Reply 2:

It is indeed possible to solve the structure, containing only three atoms (Fe, N1, N2) in the unit cell, with only 10 parameters to be refined. The data/parameter ratio 4.9 is not perfect for a standard dataset, but considering the pressure and symmetry it is quite acceptable. We didn't use any theoretical models as a starting point of the refinement. The solution was done *ab initio* using well-known crystallographic techniques. On the first step of the structure solution, the position of the heavy atom (Fe) is revealed by the solution program. Nitrogen atoms are then placed based on the analysis of the residual electron density.

The quality of the structure solution is represented by the refinement indicators, by the residual electron density and, of course, by crystal chemical consistency. A widely-accepted in the crystallographic community checkcif routine didn't have any serious complains about the solution and refinement (as may be seen from files supplemented with the revised manuscript).

We add several raw diffraction images to the supplementary materials (Supplementary Figures 1-3, Pages 2-4) showing the standard quality of grain indexing in our datasets.

We believe that the supplementary figure 3 will remove any doubts about the quality of the dataset and indexing:

Supplementary Figure 3. Reconstructed reciprocal lattice plane ($h0l$) of FeN₄ at 135 GPa.

Changes in the main text:

The section “X-ray diffraction” in “Methods” is significantly extended (Page 9).

=====

=====

Comment 3:

What is the explanation for the significant difference in bond distances for N1-N2 and N2-N2 (1.33 and 1.43 Å) if they are both considered as N-N single bonds in FeN₄? Increased electron density would certainly open up the N2-N2 distance, but 1.43 is the longest known pernitride distance! In this case, the pi* orbitals are fully occupied (N24-) and it's not clear how this could be consistent with the current result.

Reply 3:

These two bonds have different nature: N2-N2 bond is a $sp^3(N)-sp^3(N)$ bond, while N1-N2 is a $sp^2(N)-sp^3(N)$ bond. It is very well known, that the more s-character the bond has, the shorter it will be.

Regarding the distances in pernitrides:

The Reviewer, probably, refers to the distances in OsN_2 [2], Let us cite from this article: *“Because of the large mass difference between nitrogen and the parent metals, nitrogen positions could not be conclusively determined by Rietveld refinement for either of the two compounds”*.

So, we cannot exclude an uncertainty in the N-N distance of 1.43 Å reported in [2]. There is also an uncertainty in the determination of the N2-N2 bond distance in FeN_4 that we honestly provide in the manuscript: 1.43(4) Å. Therefore, we believe that there is no contradiction between these observations. Furthermore, we would like to mention several examples of similar bond distances in predicted polynitrides at similar or even higher pressures:

LiN_3 (*C2/m*) d(N-N) = 1.443 at 100 GPa [3]

BeN_4 (*P2₁/c*) d(N-N) = 1.438 at 120 GPa [4]

Mg_5N_4 (*Cmca*) d(N-N) = 1.501 at 200 GPa [5]

=====
=====

Comment 4:

Regarding the presence of infinite N chains, it is unclear, based upon the presented data and bond distances, to the Reviewer that the chains are really infinite and not the result of some unmodelled disorder or weak(er) interaction between discrete $(N_4)_2^-$ anions. The R-factor of the best data set for FeN_4 is significantly worse than the other presented data sets, $R_1 = 0.0879$, and the N atoms are not refined anisotropically while they are in the others. While this is to be expected from a low-symmetry material, the determination of the atom positions and their interactions can certainly be improved. This would help elucidate the presence of infinite chains vs discrete anions/disorder.

Reply 4:

We are grateful to the Reviewer for this comment. It stimulated us to continue searching for stronger diffracting grains and re-examining the data-sets, and we succeed eventually. The integration gave a much better R factors than before (R_{int} of 3.73 % and R_1 of 6.42 %), which we consider as a super-result for the dataset collected at almost 140 GPa. Furthermore, the residual electron density is improved to $\Delta\rho_{max}/\Delta\rho_{min}$ 0.98/-1.09 that is clearly sufficient to claim that the refinement is of a good quality. We have updated the Table 2 and Figures 2a,2b according to the new refinement, and submit an updated cif file and checkcif report for FeN_4 .

We cannot imagine any other kind of interaction between two nitrogen atoms at a distance of 1.43 Å from each other, rather than a covalent bond.

Changes in the main text:

Slightly changed lattice parameters and atomic coordinates of FeN₄ (Page 5, Figure 2, Table 2).

=====

Comment 5:

The authors point out that the best single-crystal refinement of FeN₄ was obtained at ~135 GPa, but that the phase persists on decompression down to 22.7 GPa. Although they were unable to obtain single-crystal refinements below 94.5 GPa, it should be possible to provide the lattice indexing for data collections below 94.5 GPa. This would allow for additional data points for the determination of the EOS beyond the small number presented (4).

Reply 5:

Due to the significantly reduced crystal quality, the indexing of the single-crystal datasets below 94.5 GPa was not possible. The phase identification was made based on the still X-ray diffraction images of the sample. FeN₄ has three characteristic strong peaks that do not overlap with the peaks from any other phases - (0 1 0), (0 0 1), (0 -1, 1). We followed the evolution of these peaks with pressure, however, we, of course, were not able to determine 6 lattice parameters for this P-1 phase from the d-spacings of these three peaks. Therefore, we don't discuss the volume-pressure behavior of FeN₄ down to 22.7 GPa. Of course, we may simply omit points below 94.5 GPa, however, we strongly believe that the readers must be aware of this observation. In the revised version of the manuscript we add a comment about the data below 94.5 GPa and add supplementary figure 5 (Page 6) illustrating our arguments regarding the existence of FeN₄ down to 22.7 GPa.

Changes in the main text:

The sentence in the main text (in the bottom of the Page 6) is formulated as follows: "We were able to decompress the sample containing FeN₄ obtained in the Experiment #3 down to 22.7(2) GPa (Supplementary Figure 5), however, the crystal quality significantly decreased and it was not possible to collect a single-crystal diffraction dataset suitable for the reliable indexing and structure refinement below 94.5 GPa".

=====

Comment 6:

The phase FeN was present throughout the study. What was the phase fraction of FeN during the various points in the study? Are the authors able to isolate crystalline regions of a single phase or is each other phase always mixed with FeN? If not, what methods are used to deconvolute the presence of multiple single-crystalline phases? What is the reason(s) for the large range of stability presented by FeN?

Reply 6:

We are not able to judge quantitatively about the phase fraction of FeN in our samples, but the domains can be easily separated from each other. In the revised manuscript we add several diffraction images to the supplementary materials (supplementary figures 1-3, pages 2-4) to show that the diffraction spots from different phases and domains can be easily deconvoluted, because the absolute majority of the peaks don't overlap with each other.

We do not focus much on FeN in our study, because it was already studied in three independent works [6–8]. The wide range of stability is discussed, for example in the Ref. [8].

Changes in the main text:

Added a reference to Laniel et al. [8] (Page 4).

Added a reference to the supplementary figures 1-3 (Page 9).

=====

Comment 7:

The authors seem to claim that FeN₂ is a pernitride on page 4, but the bond length and compressibility seem to rule this out. The Fe³⁺ interpretation seems plausible, but the authors might consider this as a diazenide with an elongated N=N bond due to the extra electron in the conduction band.

Reply 7:

In the revised version of the manuscript we avoid mixing terms pernitride and diazenide.

=====

Comment 8:

The authors should consider revising the sentence “High nitrogen content and stability are mutually exclusive, thereby making the synthesis of such materials challenging.” Of course, pure nitrogen phases (like cg-N) can be thermodynamically stable depending on conditions. What the authors are really referring to here is metastability.

Reply 8:

We agree that initial formulation of this statement was not strict enough. We rephrase this sentence in the abstract as follows: “In general, the higher nitrogen content leads to a higher energetic performance,

however in most cases high nitrogen content and stability at ambient conditions are mutually exclusive, thereby making the synthesis of such materials challenging.”

=====

Comment 9:

It is not clear why the authors separate chain and ring structures in the discussion of polymeric nitrogen phases in the introduction. Much progress has been made in pentazolite synthesis recently and several additional references (papers by Steele in ref. 30, for example) could be expanded.

Reply 9:

We had a short discussion regarding synthesis of cesium pentazolite salt in the introduction. In the revised version of the manuscript we add also several references to predicted structures with N5 and N6 ring structures and more complex nitrogen networks (Middle of the Page 2).

Changes in the main text:

Added 17 more references to different predicted polynitrides with various poly-nitrogen units (Page 2).

=====

Comment 10:

TiN₂ has the Al₂Cu structure, not marcasite as stated on page 3.

Reply 10:

We thank the Reviewer for pointing this mistake out. We have corrected this (Page 2).

=====

Reviewer #4 (Remarks to the Author):

This paper reports the extreme conditions synthesis of several new Fe-N systems, the authors should be congratulated on the successful application of very challenging experimental conditions resulting in novel findings.

The topic of high energy density materials is topical and attractive to a wide range of readers, both in the direct community and the wider scientific community.

The majority of the conclusions seem well founded, however certain aspects of the paper would benefit from either further explanation or investigation. Additionally there are minor typographical errors which detract from the readers flow through the paper. Concerns about the detailed methods and conclusions are outlined below with additional comments on the typographical errors later in this report.

=====

Comment 1:

From the current description on the synthesis of the materials, a noted challenging field, the details do not clarify the conditions suitably for reproduction of the experiment. The laser heating is described as happening at 50GPa at 1900(100)K, the error on the pressure here is difficult to obtain, but more pertinently the details of if this is the loading pressure or the maximum pressure obtained should be described. If this is the loading pressure then the reaction probably occurs as a pressure that is far in excess of this by impact of the extreme heating of the sample. Additionally the details of the pressure measurements are not entirely clear and the inclusion of all parameters for this, for instance: if this is a measurement from Ruby fluorescence then have the authors accounted for the lack of linearity in the pressure response due to the pressure range investigated as well as the effects of temperature?

Reply 1:

We didn't measure pressure directly during the heating of the sample. However, we have measured pressure directly before and after the laser-heating. In this study we didn't aim to construct precise phase boundaries between all Fe-N phases, and believe that this small uncertainty in pressure in no case affects the main result of the manuscript.

Changes in the main text:

Provided pressures before and after laser heating in the Table 1 (Page 22).

We add a comment that the pressure during the heating might have been different (Page 8).

=====

Comment 2:

On page 4 the FeN system is described as transforming back to the ZnS form, without detailing which ZnS motif is being mimicked.

Reply 2:

We corrected the sentence as follows: On decompression, it is stable down to ambient pressure, but with time, it transforms to zinblende-structured FeN (Page 4).

=====
=====

Comment 3:

A large uncertainty in the paper, in its current form, is the interpretation of the bonding of the N2 moieties described on page 6. Comparisons of N2 bonding distances under ambient conditions to those at 135GPa is questionable when used for the determination of multiplicity of bonding. Why not compare to data from higher pressure studies? Moreover within this section the value of 1.33 Å is described as being singly bonded (the same as 1.22Å whereas 1.43Å is described as doubly bonded), from the values given 1.33 falls directly between them. This midpoint is further backed by the calculated electronic properties, whereby a simple description of this bond as single should be strongly interrogated.

Reply 3:

There is, probably, some misunderstanding here. Bonds of lengths 1.33 Å and 1.43 Å were described as single N-N bonds, while the bond of length 1.22 Å – as double N=N bond.

Please also see the reply to the Comment#3 of the Reivewer#3 regarding the bond lengths.

Unfortunately, the information about N-N bond lengths at high pressure is very limited (one even could say that accurate experimental data simply absent), therefore, we cannot directly compare the distances at 135 GPa with some existing database.

We also would like to point out here, that the assignment of the bond order was done not only based on the bond lengths, but also based on the coordination geometry of the N-atoms. N1 atoms clearly have triangular planar coordination, while N2 atoms – tetrahedral.

=====
=====

Comment 4:

The assertion on page 7 that the compound may be recovered to ambient conditions through decompression under inert atmosphere would benefit from further discussion, the sample currently is shown to degrade at 22.7(2)GPa (note there is error inclusion here) however under those conditions the sample MUST surely still be maintained in an inert atmosphere hence the implication that the atmosphere is causing the degradation is somewhat flawed.

Reply 4:

We agree with the Reviewer, that we do not have sufficient proofs that non-inert atmosphere may cause the degradation of the sample. We exclude this statement from the revised manuscript.

Changes in the main text:

The sentence (top of the Page 7) is rephrased as follows: "Therefore, there is still a chance that FeN₄ stabilization can be potentially achieved through low-temperature decompression."

=====

=====
Comment 5:

The following relate to typographical issues with the manuscript in its current form:

Page 1: chromium carbide Cr₃N₂ is incorrect

Page 2 and possibly elsewhere referencing inconsistencies 3-5 and then 10,11,12

Page 2: paragraph 2 "as of today" reword ", as yet," ?

Page 3 DAC is not defined prior to use - very common usage but to the wider audience this may be lost

Page 3 paragraph 2 "Major challenge" word missing add either "The" or "A" at start of sentence, "rely too strongly" loose english "is reliant upon" or "relies strongly"

Page 4 1st paragraph "we were no longer observing this phase" again loose replace with "this phase was no longer observed"

Page 7 spelling of stabilization

Page 8 referencing inconsistency period after and before reference numbers

Page 10 missing space between were and performed

Reply 5:

We are very grateful to the Reviewer for such a careful inspection of the manuscript. We have corrected all mentioned typos.

References:

- [1] L. M. J. Kroon-Batenburg, J. R. Helliwell, et al., *IUCrJ* **4**, 87 (2017)
DOI:10.1107/S2052252516018315.
- [2] J. A. Montoya, A. D. Hernandez, et al., *Appl. Phys. Lett.* **90**, 2005 (2007)
DOI:10.1063/1.2430631.
- [3] D. L. V. K. Prasad, N. W. Ashcroft, et al., *J. Phys. Chem. C* **117**, 20838 (2013)
DOI:10.1021/jp405905k.
- [4] S. Zhang, Z. Zhao, et al., *J. Power Sources* **365**, 155 (2017)
DOI:10.1016/j.jpowsour.2017.08.086.
- [5] S. Yu, B. Huang, et al., *J. Phys. Chem. C* *acs.jpcc.7b00474* (2017)
DOI:10.1021/acs.jpcc.7b00474.
- [6] W. P. Clark, S. Steinberg, et al., *Angew. Chemie Int. Ed.* **56**, 7302 (2017)
DOI:10.1002/anie.201702440.
- [7] K. Niwa, T. Terabe, et al., *Inorg. Chem.* **56**, 6410 (2017) DOI:10.1021/acs.inorgchem.7b00516.
- [8] D. Laniel, A. Dewaele, et al., *J. Alloys Compd.* **733**, 53 (2018)
DOI:10.1016/j.jallcom.2017.10.267.

Reviewers' comments:

Reviewer #1 (Remarks to the Author):

The author response and revisions are acceptable in addressing my comments. The paper should be accepted for publication in the current version.

Reviewer #2

Editorial Note: this reviewer provided no further comments to the authors.

Reviewer #3 (Remarks to the Author):

Bykov et al. submit a revised version of their Fe-N paper, now including updated crystal information and a few additional details. Overall it's good work, but the major concern with the paper remains. The claim "FeN₄ is the first unambiguously experimentally confirmed nitrogen compound with polymeric nitrogen chains" is not adequately supported.

There is still no independent evidence to support the formation of FeN₄ with polymeric N chains. The Raman data unfortunately just show cg-N and huge background. The Mossbauer data are consistent with Fe²⁺, but cannot confirm any other details of the structure (It would also be prudent to show the Mossbauer of the coexisting FeN phase at similar conditions to rule out any possible spin transitions that could give a similar spectrum. So far, only FeN at 18 GPa has been reported to my knowledge).

The authors are correct that single-crystal diffraction can usually stand alone when structures are complete with high redundancy of data. But here we have a very limited data set and almost no symmetry! It would be useful for the authors to report the raw hkl file, before merging any reflections, to give readers a better picture of the data. I agree that for the given indexing, the proposed model is probably the best refinement (although the authors still don't explain how they can use anisotropic refinements for other structures, but not FeN₄. An obvious warning sign.). But we have to assume that lattice is correct. With such a low range of indices, this is simply not sufficient to stand alone for triclinic with six degrees of freedom. One could easily change the indexing, for example move one lattice vector, and still easily explain Fig S3. The unit cell could be totally wrong, which would mean the composition is wrong and all analysis of FeN₄. More observations would help the diffraction data stand alone, but too much uncertainty remains to support the current claims at present. Another form of evidence is required to support the conclusion. A modified structure would probably also give a better explanation for the N bond distances. Small note, did the authors include wavelength DISP instructions in their refinements?

The DFT calculations rely on the experimentally determined structure and therefore do not represent an independent verification. The authors should show unbiased structure searching results to make this claim. They may in fact find a better structural model this way. This would be the minimum requirement to support the current conclusions in the absence of additional experimental data. Also, it seems that only selected symmetry directions are shown for phonon dispersion relations – are all other directions stable?

A small problem: how can they obtain the tick marks for FeN₄ in Fig S5 if they cannot index this phase at lower pressure? Something is inconsistent here.

It would also be useful to compare the modulus values at the same pressures in Fig 4. If K0 for FeN₄ is not known (use theory?) then compare K94.5 (or similar) for all?

In conclusion, more evidence is needed to support the FeN₄ claim with polymeric chains. This could be a full DFT-based structure search, more complete XRD or some form of spectroscopic data. I could also support a softening of the conclusion, e.g., the data may be consistent with the first polymeric structure, but this approach would take away most of the appeal for the target journal.

Reviewer #4 (Remarks to the Author):

The authors have included a full spectrum of corrections and the paper has been significantly enhanced by virtue of this.

Reply to the Reviewer's comment:

"Bykov et al. submit a revised version of their Fe-N paper, now including updated crystal information and a few additional details. Overall it's good work, but the major concern with the paper remains. The claim "FeN₄ is the first unambiguously experimentally confirmed nitrogen compound with polymeric nitrogen chains" is not adequately supported.

There is still no independent evidence to support the formation of FeN₄ with polymeric N chains. The Raman data unfortunately just show cg-N and huge background. The Mossbauer data are consistent with Fe²⁺, but cannot confirm any other details of the structure (It would also be prudent to show the Mossbauer of the coexisting FeN phase at similar conditions to rule out any possible spin transitions that could give a similar spectrum. So far, only FeN at 18 GPa has been reported to my knowledge)".

Reply:

Single-crystal X-ray diffraction provides sufficient evidence for the existence of FeN₄. Single-crystal X-ray diffraction is, probably, the best method for studying new compounds as evidenced by ~900 thousand crystal structures deposited in the CCDC, 150 thousand structures in ICSD, 140 thousand structures in PDB. Other methods may indeed give important information about the properties of the material or just support the XRD, but it is not needed, as established by decades of development of modern crystallography and inorganic chemistry. In our manuscript, we make the conclusions based only on the structure determination. As suggested by referees in the previous review round, we included Raman and Mössbauer spectroscopy data to demonstrate that they do not contradict the interpretation of the structural result. However, we clearly state that we do not discuss or draw any conclusions based on the results of spectroscopy.

"The authors are correct that single-crystal diffraction can usually stand alone when structures are complete with high redundancy of data. But here we have a very limited data set and almost no symmetry! It would be useful for the authors to report the raw hkl file, before merging any reflections, to give readers a better picture of the data".

Reply:

We add a raw *.hkl* file as a supplementary file in the revised version of the manuscript. We would like to draw attention of Reviewer #3 to the number of papers, in which much more complex crystal structures (with triclinic symmetries, or with incommensurate crystals structures) were determined using single-crystal XRD even though the datasets were not complete:

Merlini M. et al. (2012). Structures of dolomite at ultrahigh pressure and their influence on the deep carbon cycle. *Proceedings of the National Academy of Sciences of the United States of America*, 109(34), 13509–13514.

Merlini M. et al. (2012). CaCO₃-III and CaCO₃-VI, high-pressure polymorphs of calcite: Possible host structures for carbon in the Earth's mantle. *Earth and Planetary Science Letters*, 333, 265–271.

Arakcheeva A. et al. (2017). Incommensurate atomic density waves in the high-pressure IVb phase of barium. *IUCrJ*, 4(2), 152–157.

"I agree that for the given indexing, the proposed model is probably the best refinement (although the authors still don't explain how they can use anisotropic refinements for other structures, but not FeN₄. An obvious warning sign)".

Reply:

We have used anisotropic refinements for all other structures because there was a sufficient data/parameter ratio for those datasets. The usage of anisotropic refinement was justified by a significant drop of the R₁ value. In other words, the anisotropic refinement was justified by a Hamilton test. Inclusion of

the anisotropic refinement for FeN₄ would lead to a small data/parameter ratio and to over-interpretation of the data. We are following the widely accepted crystallographic policies. Contrary to that suggested by Referee #3, it would be a “warning sign” if we did use anisotropic refinement here.

“But we have to assume that lattice is correct. With such a low range of indices, this is simply not sufficient to stand alone for triclinic with six degrees of freedom. One could easily change the indexing, for example move one lattice vector, and still easily explain Fig S3. The unit cell could be totally wrong, which would mean the composition is wrong and all analysis of FeN₄. More observations would help the diffraction data stand alone, but too much uncertainty remains to support the current claims at present. Another form of evidence is required to support the conclusion. A modified structure would probably also give a better explanation for the N bond distances”.

Reply:

We have used 82 reflections of FeN₄ collected up to *d*-shell of 0.75 Å for the determination of 6 lattice parameters. We are fully convinced that this number and resolution are sufficient for unambiguous unit cell determination. Unit cell determination was done with the standard routine DIRAX included in *Crysalis^{Pro}* software package. The script finds the unit cells for the given set of the reflections and then selects the most symmetric cell with the smallest volume. Reciprocal space reconstructions show that no additional reflections are present, i.e. the unit cell parameters are not doubled/tripled and so on. Thus, there is no reason to think that our structural model might be deficient. It is not clear what the Reviewer meant under “move one lattice vector and still easily explain FigS3”. If the Reviewer meant the change of the unit cell setting, then this procedure would not change the structural model. Needless to say that the agreement factor of our model is convincingly low: R₁ = 6.42 %.

“Small note, did the authors include wavelength DISP instructions in their refinements?”

Reply:

For the refinement we use the computer program Jana2006, that uses appropriate energy-dependencies of the atomic form factors from the International tables for Crystallography Vol C. Chapter 6

“The DFT calculations rely on the experimentally determined structure and therefore do not represent an independent verification”.

Reply:

This is incorrect. Giving the complexity of the crystal structure, the fact that fully relaxed structure does not differ from the original input, when the forces on all the atoms, as well as stresses (except pressure) are zero, provides very strong independent support that the structure is at least metastable at this pressure.

“The authors should show unbiased structure searching results to make this claim. They may in fact find a better structural model this way. This would be the minimum requirement to support the current conclusions in the absence of additional experimental data”.

Reply:

This is incorrect. A crystal structure prediction is totally meaningless in the context of this work. In fact, a prediction cannot confirm the structure; the number of successful structure predictions is known to be much less than the total number of “predictions” made. Numerous failures of the approach are well known in the field, though they are poorly documented, because wrong “predictions” are not published.

“Also, it seems that only selected symmetry directions are shown for phonon dispersion relations – are all other directions stable?”

Reply:

It is the common practice to plot the phonon dispersion relations along the high-symmetry directions, and we follow it in the present work. To confirm that there are no imaginary frequencies in a general k-point, we complement the figure 3 with the calculated phonon density of states. One can see that it is finite only for positive frequencies, confirming our earlier conclusion on the dynamical stability of FeN₄.

“A small problem: how can they obtain the tick marks for FeN₄ in Fig S5 if they cannot index this phase at lower pressure? Something is inconsistent here.”

Reply:

Only first three powder peaks of FeN₄ do not overlap with the peaks from other phases, the rest do overlap. Therefore, the refinement of 6 lattice parameters might be insufficiently accurate. For this reason, we are simply careful with our conclusions and don't want to include the numerical data here. But we want to show the readers that the phase exists down to very low pressures.

“It would also be useful to compare the modulus values at the same pressures in Fig 4. If K₀ for FeN₄ is not known (use theory?) then compare K_{94.5} (or similar) for all?”

Reply:

We compared the bulk moduli of FeN and FeN₄ at 94.5 GPa and added the following sentence on Page 6 of the manuscript: “The bulk modulus of FeN₄ at 94.5 GPa is slightly smaller than that of FeN (K_{94.5}(FeN) = 679 GPa)”. We do not think that the extrapolation of the bulk modulus of FeN₂ up to 95 GPa is justified.

“In conclusion, more evidence is needed to support the FeN₄ claim with polymeric chains. This could be a full DFT-based structure search, more complete XRD or some form of spectroscopic data. I could also support a softening of the conclusion, e.g., the data may be consistent with the first polymeric structure, but this approach would take away most of the appeal for the target journal.”

Reply:

We are fully convinced that no further evidences are needed and other three Referees agree with us.

Reviewers' comments:

Reviewer #3 (Remarks to the Author):

The authors are reluctant to make any changes or provide additional data to support their manuscript. I have performed independent calculations and reanalyzed their data to show an alternative structural model with a better fit and lower energy. The authors need to revise their work.

Firstly, the authors argue that hundreds of thousands of structures have been solved with SCXRD. While this is true, it is not fair to compare thousands of structures at 1 atm to high pressure with sparse data. This is a non sequitur. The Raman and Mossbauer provided simply do not satisfy the reviewer's comments for additional evidence.

The authors point out several papers that solved complex structures with high-pressure data. I agree that it is possible to obtain information, but we should be fair and realistic. They compare data with reciprocal space coverage of 50% to the same resolution whereas the current paper is only 34%. I commend the work in the barium paper, but it is unclear how this can support the case here – one of the key conclusions is that more complete data are needed to fully understand the structure! While complex structures can indeed be solved with incomplete data, the list of incorrect structures is much, much longer! That's why crystallography journals have strict requirements on the data. Hence, more information is needed in the absence of a complete solution!

The author's calculation simply makes local adjustments to the input parameters in order to minimize forces and therefore cannot represent an independent verification; this claim must be discarded. It can only indicate that the structure is possible or "at least metastable" as the authors say in their reply. An unbiased confirmation of the proposed structure would not be "meaningless." If correct, it would, in fact, support the conclusion very much. To help be constructive, I have therefore performed the calculations for FeN₄ and do not find their proposed structure to be energetically competitive. I do find another P-1 structure that can potentially describe these data, but the assumption of composition remains.

As I suspected, it is likely that other indexing solutions are plausible for all the collected reflections. Most indexing software presents a list of possible solutions that the user must then select and improve by removing reflections. The authors did not provide sufficient details on their indexing with multiple phases present and never showed convincingly that their chosen indexing is the most likely. Once I re-index their data based on the independent model from ab initio searching, I obtain a structure that gives nearly identical diffraction. For simplicity, I compare the powder diffraction patterns of the two structures below.

The new lattice parameters that I propose are: $a=3.54$ $b=4.24$ $c=4.40$ $\alpha=108.7$ $\beta=90.7$ $\gamma=114.5$. The authors should easily find the atomic positions and will calculate a significantly lower energy for this structure than the one they propose. I did not perform the full structural refinement because

I did not have time to revise the hkl file, but I ran the two structures through the program Endeavor and the larger unit cell reduces the R-factor by ~4%.

If the authors make the claim of the first crystalline structure with polymeric nitrogen chains, the burden of proof is on them. In the absence of a complete crystal structure, it is not unreasonable to request additional evidence to support the claim. Here, I show that unbiased structure searching can be very helpful for independent verification. It turns out that the structure the authors propose is similar to the one I found – the main difference being the cell size (I suspect that the bonding distances will also improve, but errors will still be large). The authors should be cautious with putting too much faith in their incomplete data.

The final question that remains is whether the composition is correct. Given that that multiple indexing solutions can describe the limited data set, it may be possible that the composition is wrong. Even though the structure searching solution is similar, I still made the assumption of FeN₄. It is a good sign that the independent results agree now, but another variation cannot be dismissed entirely without another form of evidence. I suggest they add some discussion to this point in the paper.

Reply to Reviewer #3

“The authors are reluctant to make any changes or provide additional data to support their manuscript.”

The Referee is not objective. We not only substantially improved the manuscript, but also provided new data in previous review rounds.

“I have performed independent calculations and reanalyzed their data to show an alternative structural model with a better fit and lower energy. The authors need to revise their work.”

We are grateful to Reviewer #3 for the detailed inspection of our data, but the “*alternative structural model with a better fit and lower energy*” is incorrect. We have reanalyzed Reviewer’s “*alternative structural model*” and found that Reviewer #3 simply provided us with a typical example of how a combination of **powder** XRD and *ab initio* structure prediction may lead to wrong results.

Indeed, Reviewer #3, writes that he/she re-indexed our XRD data, and found (independently) another structure solution, which describes well the **powder pattern**. By reducing the single-crystal dataset to the powder one, the Reviewer lost a lot of information, as powder XRD data may give only the lengths of the reciprocal space vectors, whereas single-crystal diffraction also provides information about their orientation, thus, making determination of the space group and indexing unambiguous.

The Reviewer suggests a unit cell with the parameters: $a = 3.54$, $b = 4.24$, $c = 4.40$ Å, $\alpha=108.7$, $\beta=90.7$, $\gamma=114.5^\circ$. This unit cell, in fact, can be obtained from our cell ($a = 2.5089$, $b = 3.5245$, $c = 3.5409$ Å, $\alpha = 105.08$, $\beta = 110.260$, $\gamma = 92.03^\circ$) by a simple transformation:

$$\begin{pmatrix} \mathbf{a}_r \\ \mathbf{b}_r \\ \mathbf{c}_r \end{pmatrix} = \begin{pmatrix} 0 & 0 & -1 \\ 1 & 1 & 1 \\ 1 & -1 & 0 \end{pmatrix} \begin{pmatrix} \mathbf{a} \\ \mathbf{b} \\ \mathbf{c} \end{pmatrix}$$

and *vice versa*

$$\begin{pmatrix} \mathbf{a} \\ \mathbf{b} \\ \mathbf{c} \end{pmatrix} = \begin{pmatrix} 0.5 & 0.5 & 0.5 \\ 0.5 & 0.5 & -0.5 \\ -1 & 0 & 0 \end{pmatrix} \begin{pmatrix} \mathbf{a}_r \\ \mathbf{b}_r \\ \mathbf{c}_r \end{pmatrix}$$

where \mathbf{a}_r , \mathbf{b}_r , \mathbf{c}_r the Reviewer’s lattice vectors, and \mathbf{a} , \mathbf{b} , \mathbf{c} – ours.

The lattice transformations cannot change the structure model. We have already had this discussion during the previous round of the review process and would like to underline once again that “different indexing solution” does not mean “different structure model” (see **Figure 1** below). The transformation (as above) applied to our cell leads to the *I*-centered triclinic unit cell, and, although the Reviewer claims that his/her structure has the symmetry $P\bar{1}$, **Figure 2** (see below) shows that the unit cell in the Reviewer’s setting is definitely *I*-centered. This result is unambiguous and could not be affected by a limited data coverage at high pressure. It disproves the Reviewer’s “*alternative structural model*” as inconsistent with our experimental data.

Additionally, any comparison of R-factors calculated in the Endeavor program for “synthetic powder diffraction” data generated based on the unit cells containing different amount of formula units (and thus different number of refined parameters) has a little meaning.

In the lights of the above analysis of the structural model proposed by the Reviewer, further discussion is probably not necessary. However, following the requirements of the point-by-point reply, we address below all Reviewer’s comments.

“The Raman and Mossbauer provided simply do not satisfy the reviewer’s comments for additional evidence.”

From the very beginning, we explained in our communications with the reviewers that Mössbauer and Raman data are not only very difficult to acquire in our case, but they may only support X-ray diffraction results, and would not play any definitive role. The Reviewer requested to present these data and we applied efforts to collect them. In such circumstances the Reviewer rather might acknowledge that we were right and these spectroscopy data were indeed not really necessary, than to express his/her non-satisfaction in such an ungrateful way.

“The author’s calculation simply makes local adjustments to the input parameters in order to minimize forces and therefore cannot represent an independent verification; this claim must be discarded. It can only indicate that the structure is possible or “at least metastable” as the authors say in their reply.”

This is incorrect; instead of “local adjustments to the input parameters in order to minimize forces”, we made the full structural optimization and thus our theoretical calculations support static and dynamic stability of the atomic configuration revealed by X-ray diffraction data.

“An unbiased confirmation of the proposed structure would not be “meaningless.” If correct, it would, in fact, support the conclusion very much. To help be constructive, I have therefore performed the calculations for FeN₄ and do not find their proposed structure to be energetically competitive. I do find another P-1 structure that can potentially describe these data, but the assumption of composition remains.”

We have already addressed this question above and found the model proposed by the Reviewer to be incompatible with our experimental observations.

“As I suspected, it is likely that other indexing solutions are plausible for all the collected reflections. Most indexing software presents a list of possible solutions that the user must then select and improve by removing reflections. The authors did not provide sufficient details on their indexing with multiple phases present and never showed convincingly that their chosen indexing is the most likely.”

The indexing was done using the program Ewald^{Pro} - the reciprocal space viewing tool for the CrysAlis^{Pro}. The procedure for the analysis of multocrystal and multiphase samples is demonstrated in the User’s manual. This program is used by thousands of researchers all over the world. We have already demonstrated the quality of the indexing and of the crystal in the Supplementary Figures 2 and 3. In the revised version of the manuscript we have explicitly added a reference to the Ewald^{Pro}.

[Redacted]

Figure 2. Reconstruction of the reciprocal lattice planes $(h-2l)$ (top) and $(h-3l)$ (bottom) for the FeN_4 phase using the Reviewer's unit cell. Reflections obey an extinction rule $h + k + l = 2n$. Red circles indicate the positions of absent reflections. Additional features on the reconstructions are produced by the diffraction from different grains of the same phase, diamond and nitrogen.

REVIEWERS' COMMENTS:

Reviewer #4 (Remarks to the Author):

It is clear that the authors and reviewer #3 have reached something of an impasse. This basically relates to the weighting the authors put on the structural refinement of the F3N4 polymeric system. Both sides have significant strength in their arguments, as indeed it would be far more reassuring if the structural data were more complete, or that the refinement could be taken through to a higher level (such as including adp's) however to do this without the required data is fraught with danger. I assume that the additional reflections from different domains have already been attempted to be integrated and merged in to the final results to aid the completeness. The arguments put forward by the authors for their interpretation of the structural model from the data collected are, in my view, generally sound, however this does not remove the concerns of reviewer #3. I do not believe that the authors will be able to satisfy reviewer #3 without significant repetition of experiments in the hope (and that is all it would be) of attaining better quality data. At this point I recommend that the authors make a minor modification to the paper, adding something to their discussion of this structure as follows: "This model represents the authors' optimum refinement of the available X-ray data, which although incomplete is consistent with all other findings as indicated. Due to the reduced data to parameter ratio available, and the obvious presence of additional diffraction data from other domains, the full diffraction data are included as supplementary information; to be available for alternative interpretations if improvements in data processing become available". I believe this, or similar wording, provides the authors with sufficient strength to their document that the impact is not lost but importantly provides the community with what is obviously a challenging dataset for software development engineers. The confidence of the authors' interpretation of their data will then be indicated by their willingness to release the raw data i.e. not just the indexed and processed hkl files. This approach will also hopefully enable reviewer #3 to have confidence in the approaches taken, or indeed work on the data further after publication to expand the understanding of this interesting system, either in conjunction with the authors or as independent work.

Reviewer #4 (Remarks to the Author):

It is clear that the authors and reviewer #3 have reached something of an impasse. This basically relates to the weighting the authors put on the structural refinement of the FeN₄ polymeric system. Both sides have significant strength in their arguments, as indeed it would be far more reassuring if the structural data were more complete, or that the refinement could be taken through to a higher level (such as including adp's) however to do this without the required data is fraught with danger. I assume that the additional reflections from different domains have already been attempted to be integrated and merged in to the final results to aid the completeness. The arguments put forward by the authors for their interpretation of the structural model from the data collected are, in my view, generally sound, however this does not remove the concerns of reviewer #3. I do not believe that the authors will be able to satisfy reviewer #3 without significant repetition of experiments in the hope (and that is all it would be) of attaining better quality data. At this point I recommend that the authors make a minor modification to the paper, adding something to their discussion of this structure as follows: "This model represents the authors' optimum refinement of the available X-ray data, which although incomplete is consistent with all other findings as indicated. Due to the reduced data to parameter ratio available, and the obvious presence of additional diffraction data from other domains, the full diffraction data are included as supplementary information; to be available for alternative interpretations if improvements in data processing become available". I believe this, or similar wording, provides the authors with sufficient strength to their document that the impact is not lost but importantly provides the community with what is obviously a challenging dataset for software development engineers. The confidence of the authors' interpretation of their data will then be indicated by their willingness to release the raw data i.e. not just the indexed and processed hkl files. This approach will also hopefully enable reviewer #3 to have confidence in the approaches taken, or indeed work on the data further after publication to expand the understanding of this interesting system, either in conjunction with the authors or as independent work.

Reply:

We would like to thank the Reviewer for his/her comment. Following the Reviewer's suggestions, we have made the following amendments in the final version of our manuscript:

Page 9: The following statement was added:

"The obtained models represent the authors' optimum refinement of the available X-ray data. The full diffraction data are made available online (see Data Availability section). Their complexity for processing is obvious due to the reduced data to parameter ratio, which is characteristic for all single-crystal diffraction data sets obtained in a DAC, and due to the presence of diffraction from numerous domains. If the improvements in data processing become available, one can use the present data for reevaluation."

Page 10: The following statement was added: "Single-crystal X-ray diffraction dataset for FeN₄ at 135 GPa has been deposited to Figshare (<https://figshare.com/>) with the accession link: <https://doi.org/10.6084/m9.figshare.6471092.v1>".

We have uploaded the most important dataset (FeN₄ at 135 GPa) and made it publically available. All other datasets will be available from authors on request.